# Analysis of the Dissolution of CH$_4$/CO$_2$-Mixtures into Liquid Water and the Subsequent Hydrate Formation via In Situ Raman Spectroscopy

**Zheng Li** [1,2] , **Christine C. Holzammer** [1,3] and **Andreas S. Braeuer** [1,*]

[1] Institute of Thermal-, Environmental-, and Resources' Process Engineering (ITUN), Technische Universität Bergakademie Freiberg (TUBAF), 09599 Freiberg, Germany; zheng.li@cqu.edu.cn (Z.L.); Holzammer.Christine@gmail.com (C.C.H.)

[2] School of Energy and Power Engineering, Chongqing University, Chongqing 400044, China

[3] Erlangen Graduate School in Advanced Optical Technologies (SAOT), Friedrich-Alexander-Universität Erlangen-Nürnberg (FAU), Paul-Gordan-Str. 6, 91052 Erlangen, Germany

[*] Correspondence: Andreas.Braeuer@tu-freiberg.de; Tel.: +49-3731-392232

**Abstract:** We report an experimental study for the investigation into the suitability of hydrate formation processes for the purification of methane (CH$_4$) from carbon dioxide (CO$_2$) at a sub-cooling temperature of 6 K and a pressure of 4 MPa. The experiments were conducted in a stirred batch reactor. Three different initial CH$_4$/CO$_2$ mixtures with methane fractions of 70.1 mol%, 50.3 mol%, and 28.5 mol% were tested. The separation efficiency was quantified by measuring in situ via Raman spectroscopy the ratios of CH$_4$/CO$_2$ in the gas mixture, the liquid water-rich phase before hydrate formation, and the solid hydrate phase after the onset of the hydrate formation. The results indicated that the main separation effect is obtained due to the preferential dissolution of CO$_2$ into the liquid water-rich phase before the onset of the hydrate formation.

**Keywords:** gas hydrates; *in situ* Raman spectroscopy; molar ratio; carbon dioxide; methane

## 1. Introduction

Gas hydrates are solid crystalline compounds consisting of hydrogen bonded water networks in which guest molecules such as methane (CH$_4$) and carbon dioxide (CO$_2$) can be incorporated [1,2]. The guest molecules thermodynamically stabilize the cage-like hydrate crystals. Gas hydrates form at high pressures and low temperatures, with three common hydrate structures existing in nature, known as sI, sII, and sH [3,4]. Natural gas hydrates represent a huge energy resource [5,6]. Synthetic gas hydrates can be utilized for gas storage, gas transportation [7,8], desalination [9,10], gas separation [11–13], etc. Nonetheless, in the case of gas conveyance through pipelines the formation of gas hydrates must be inhibited in order to prevent blockage [14,15].

The formation of gas hydrates from CH$_4$/CO$_2$-mixtures has been extensively studied over the past three decades [16–19]. Some works report on the kinetics and selectivity of the gas hydrate formation process [20,21]. In some studies, liquid nitrogen [22] or coolant circulation [23] combined with gas chromatography were utilized for measuring gas compositions in a solid phase. Existing studies usually neglect the amount of CH$_4$ and CO$_2$ dissolved in the liquid water-rich phase, which after hydrate formation usually coexists with the solid hydrate phase. Holzammer et al. found that synthetically formed gas hydrates are rather gels or slurries that contain significant inclusions of a liquid water-rich phase. The fraction of the pure solid hydrate in these hydrate slurries or gels was found to be less than 30 wt.-% [24,25]. Therefore, the assumption that all the gas which is missing in the gaseous phase after the hydrate formation must have been incorporated in the formed pure solid

hydrate phase is prone to error, especially if the solubility of the gas species in the liquid water-rich phase, which coexists with the hydrate phase, is high. To the knowledge of the authors, there has been no study that correlates the composition of the $CH_4/CO_2$ gas mixture, with the dissolution of $CH_4$ and $CO_2$ into the liquid water-rich phase before the hydrate formation and the subsequent incorporation of $CH_4$ and $CO_2$ into the solid gas hydrate phase.

Raman spectroscopy is one possible method for the remote and *in situ* measurement of the composition of fluid and solid mixtures and has been applied to the analysis of gas hydrates and fluid mixtures in the context of gas hydrates [26,27].

Therefore, the object of this work is to put into context the composition of the gaseous $CH_4/CO_2$-mixture with the dissolution of $CH_4$ and $CO_2$ into the liquid water-rich phase before hydrate formation and the incorporation of $CH_4$ and $CO_2$ into the solid pure hydrate phase using Raman spectroscopy. Three different initial gas compositions were employed to evaluate the effect of the initial gas composition on the fractions of $CH_4$ and $CO_2$ in the vapor phase, the liquid water-rich phase, and the hydrate phase. Finally, $CO_2$ selectivities relevant for gas separation or gas purification processes are discussed.

## 2. Materials and Methods

### 2.1. Materials

The experiments were conducted with deionized water with a conductivity of less than 10 μS/cm and with gaseous premixed binary $CH_4/CO_2$-mixtures (Linde, molar purity 99.5%) with three certificated $CH_4$ molar fractions of 0.701, 0.503, and 0.285.

### 2.2. Apparatus

A schematic diagram of the experimental setup is given in Figure 1, which consists of a high-pressure view cell (maximum pressure 25 MPa, internal and non-adjustable volume 26 mL) and a Raman probe.

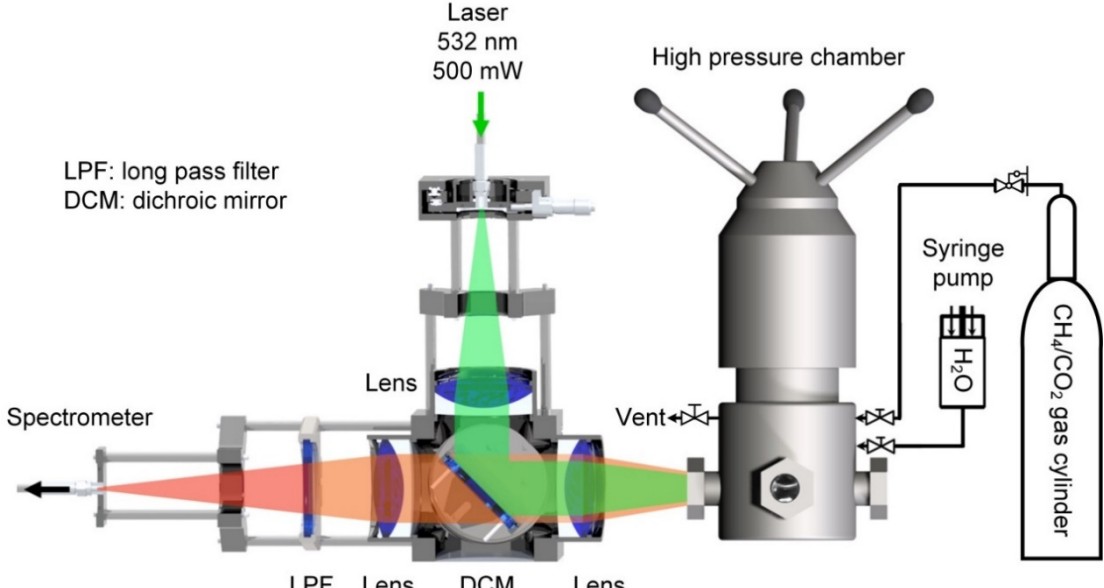

**Figure 1.** Sketch of the experimental setup, including the high pressure view cell.

The high-pressure view cell is equipped with four optical windows. Two of them are required for launching the laser through the view cell, which is required for the Raman spectroscopy and for the detection of the Raman signals. The two remaining windows were utilized for the visual observation

of the fluids and solids (gels or slurries) inside the view cell. The temperature of the cell was controlled by a cryo-compact Julabo circulator (model CF41) which pumps conditioned coolant through the double-walled cooling jacket surrounding the view cell. A PT100 temperature sensor (resolution 0.05 K, uncertainty 0.12 K) and a pressure sensor (model PAA-33X, precision 0.01 MPa) from Keller were utilized for monitoring the temperature and pressure in the cell, respectively. The temperature and pressure data were recorded and saved each second by the computer. A high precision Teledyne Isco syringe pump (model 260D) fed well-defined amounts of distilled water into the cell.

A frequency-doubled Nd:YAG laser (model 532-250-AC) from CNI with an emission wavelength of 532.27 nm and a power of 500 mW was utilized as the light source within the Raman spectrometer. The laser beam was guided by an optical fiber to the Raman probe. After collimation by a convex lens, the beam reached a dichroitic mirror (DCM) which reflects light with wavelengths shorter than 533 nm and transmits light with wavelengths longer than 544 nm (the Cut-Off wavelength is in between). Subsequently, the reflected laser light was focused into the view cell through a convex lens (focal length 100 mm). The probe volume can be assumed to have a cylindrical shape approximately 5 mm in length and 0.2 mm in diameter. The laser light was scattered elastically and inelastically from the matter within the probe volume. The inelastically Raman Stokes-scattered light (scattered at wavelengths longer than the laser wavelength) was collected by the lens in the back-scattering direction [28], collimated, and passed straightly through the dichroitic mirror, and then was guided via a lens and an optical fiber toward a spectrometer (QE65 Pro, entrance slit 100 μm, Ocean Optics with a resolution of ~15 cm$^{-1}$). A long-pass filter (LPF) was set behind the dichroitic mirror to remove any remaining elastically scattered light. The signal integration time of each Raman spectrum was 1 s.

### 2.3. Procedures

All the experiments were performed in a batch manner (with a constant volume of 26 mL). Before each experiment the inner volume of the cell was cleaned and dried. Then, 14 mL of distilled water were injected into the cell through the syringe pump. The air remaining inside the chamber was removed by purging the cell five times with the premixed feed gas mixture composed of $CO_2$ and $CH_4$, still without pressurizing the view cell. The gas inlet and exit valves were situated above the meniscus of liquid water. Then, the cell was closed and cooled to the desired experimental temperature $T_{exp}$. All the experiments were conducted at a constant initial sub-cooling of 6 K. The sub-cooling (ΔT) for hydrate formation is defined as the temperature difference between the equilibrium temperature ($T_{eq}$) and the experimental temperature ($T_{exp}$) at a specific pressure (here 4 MPa). $T_{eq}$ was predicted by CSMHyd program [29]. Thus, $T_{exp}$ for initial $CH_4$/$CO_2$-mixtures with $CH_4$ molar fractions of 0.701, 0.503, and 0.285 is 274.2 K, 275.3 K, and 276.1 K, respectively. Once the fluids inside the view cell reached the set temperature, the cell was pressurised to a fixed initial pressure $p_{ini}$ of 4 MPa by introducing the premixed $CH_4$/$CO_2$-mixture via a pressure controller slowly from the gas cylinder. The pressurization took less than 10 s. Once the pressure was 4 MPa (instant "time zero" $t_0$), the gas inlet valve was closed and a magnetic stirrer started to rotate a magnetic stirring bar situated inside the cell at its bottom. The magnetic bar rotated with ~400 rotations per minute and accelerated the dissolution of $CH_4$ and $CO_2$ from the gas phase into the water-rich phase. The pressure inside the view cell dropped due to the dissolution. The beginning of hydrate formation was indicated by another pressure drop, as additional $CH_4$ and $CO_2$ were then removed from the gas phase. At "time zero", we also started to acquire Raman spectra with a repetition rate of 3 spectra per minute and a signal integration time of 1 s per spectrum. Considering the operational conditions, the molar ratio $n_{H2O}/(n_{CO2} + n_{CH4})$ fed into the cell is larger than 29. In studies of sI hydrate structures, this ratio is typically close to 6 [30,31]. We chose a rather large ratio in order to obtain a significant change of the composition in the gas phase coexisting with the hydrate slurry/gel.

## 3. Results

Figure 2 shows as solid line the temporal evolution of the pressure inside the view cell at 276.1 K for the fed gas mixture, with a methane molar fraction of 0.285. The two grey dashed pressure curves represent two repetitions of the same experiment. The pressure curves start at "time zero" $t_0$ at 4 MPa. Between $t_0$ and $t_1$ (instant $t_1$ only labelled for the solid black curve), the pressure drops due to the dissolution of $CH_4$ and $CO_2$ into the liquid water-rich phase. This period is denoted as the gas dissolution period. At instant $t_1$, the pressure curve features a kink indicating the start of hydrate formation. Due to the exothermic nature of the hydrate formation, this pressure kink is accompanied by a temperature spike of approximately 0.5 K towards higher temperatures. Subsequently, the pressure decreases further owing to the growth of the hydrate phase which consumes further gas. The experiment is kept running until the pressure decrease is less than 0.01 MPa per hour. The period between the onset of the hydrate formation $t_1$ and the end $t_2$ of the experiment ($t_2 - t_1 \geq 20$ h) is called the hydrate formation period. This does not imply that no gas dissolves into the liquid water-rich phase during the period of hydrate formation. The pressure 2.13 MPa reached at $t_2$ is close to the pressure of 2.2 MPa expected for thermodynamic equilibrium according to the software CSMHyd, developed by the hydrate research center of the Colorado School of Mines. The hydrate layer formed at the interface to the gaseous phase acts as mass transfer resistance and can kinetically limit further hydrate growth and, along with this, also limit the pressure drop towards $t_2$. From the pressure decrease in the dissolution period (before $t_1$) and in the hydrate formation period (after $t_1$), one can already conclude that the dissolution period alone extracts a significant amount of $CH_4$ and/or $CO_2$ from the vapor phase. At the end of the experiment, the cell is depressurized, emptied, and cleaned for the next experiment. The experiments are repeated with the aforementioned procedure a minimum of three times for each initial composition of the $CH_4/CO_2$ mixture. Figure 2 also shows that the position of the Raman probe volume (laser focus beam waist) is fixed either in the vapor phase or just below the gas/liquid or gas/hydrate gel interface. Unfortunately, as there was only one Raman probe available, we were not able to simultaneously make Raman measurements above and below the interface. Therefore, serial experiments had to be conducted to obtain the Raman spectra from different phases at the same experimental condition.

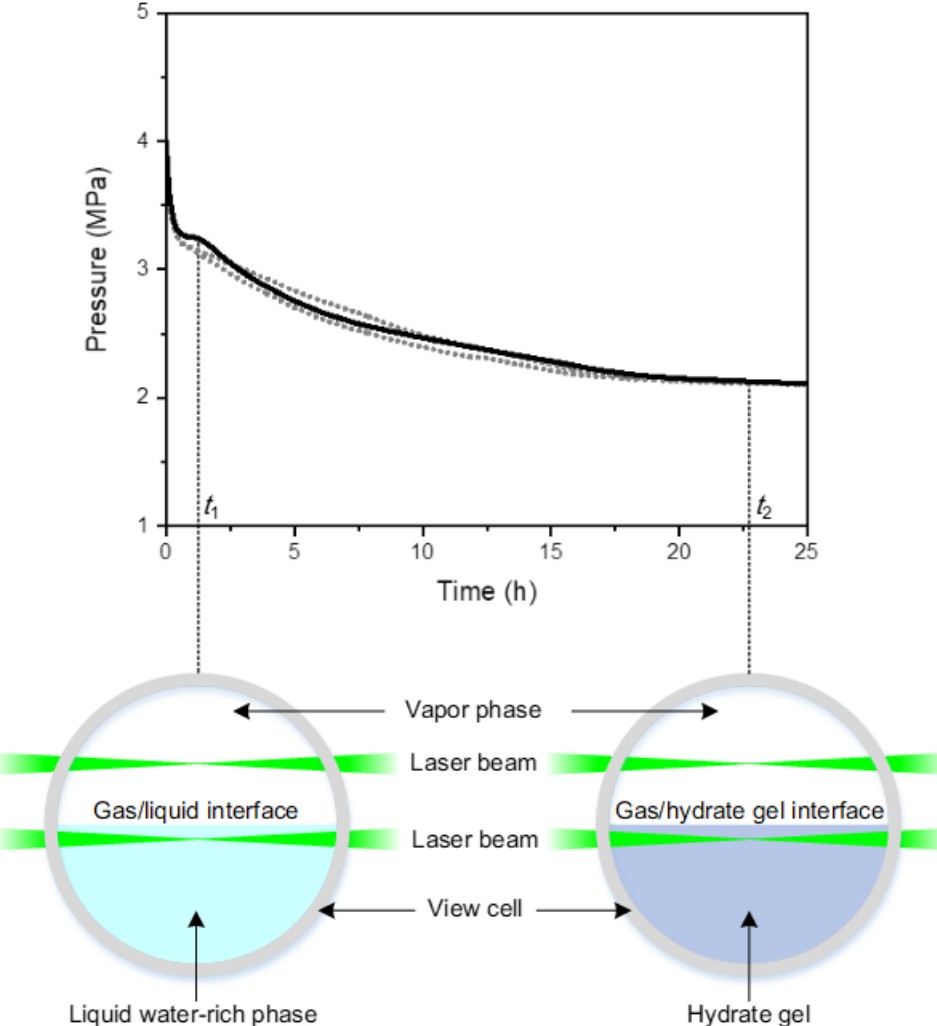

**Figure 2.** The temporal evolution of the pressure inside the view cell before and during the hydrate formation. The solid black line pressure profile represents an independent experiment, in which $t_1$ denotes the onset of the hydrate formation and $t_2$ the end of the hydrate formation period. The two grey dashed pressure curves represent two repetitions of the same experiment. Sketches are provided for two instants inside the view cell. The green lines indicate the path of the laser beam for composition measurements in the gaseous, liquid, and gel phases. Temperature T = 276.1 K.

### 3.1. Raman Spectra of CH$_4$/CO$_2$-Mixtures in Vapor Phase, Liquid Water-rich Phase, and Hydrate Gel Phase

Figure 3 shows typical Raman spectra of pure water before pressurization, the liquid water-rich phase after pressurization but before the hydrate formation, hydrate gel at the end of the experiment ($t_2$), and the vapor phase at $t_0$, all at 276.1 K. The CH$_4$ molar fraction in the feed gas mixture is 0.285. These Raman spectra are baseline corrected and normalized to their maximum.

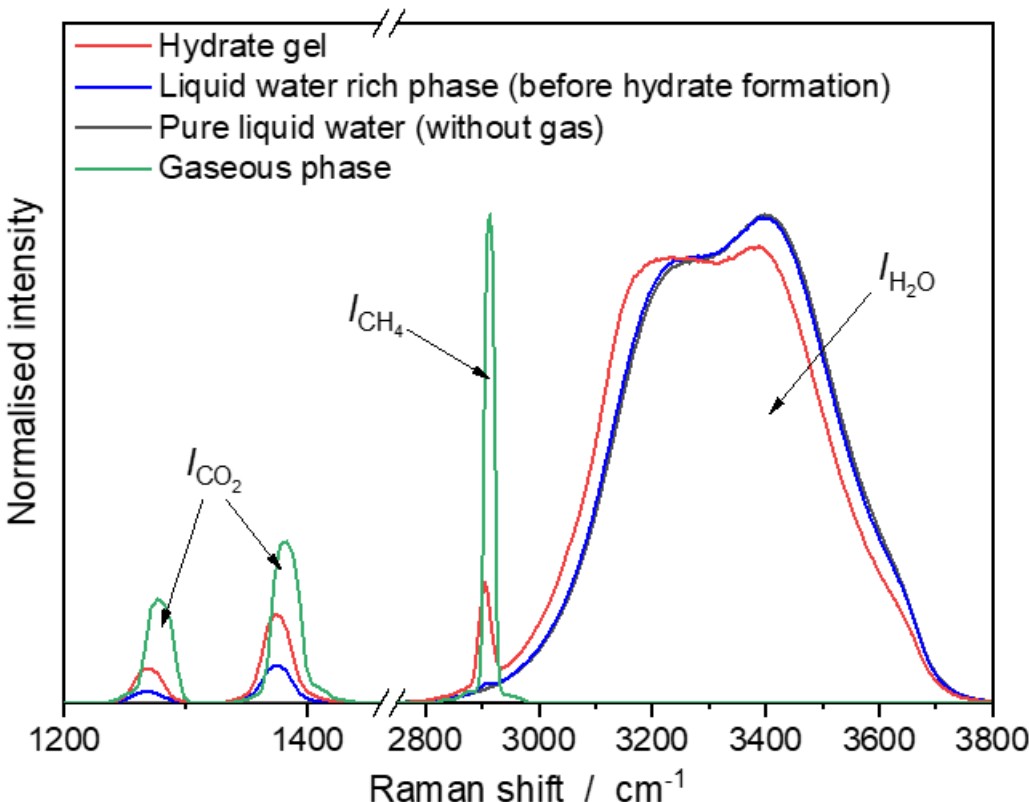

**Figure 3.** Example Raman spectra of the liquid water-rich phase at instant $t_1$ (see Figure 2), the hydrate gel at instant $t_2$ (see Figure 2), the gaseous $CH_4/CO_2$-mixture at instant $t_0$ (see Figure 2), and pure liquid water at $T = 276.1$ K. (For the pure liquid water spectrum, the pressure is $p = 0.1$ MPa).

The double peak structure, the so-called Fermi Dyad [32], at 1280 cm$^{-1}$ and 1380 cm$^{-1}$, is assignable to the $CO_2$ species (additional $CO_2$ peaks merged with the $CO_2$ Fermi Dyad, due to the resolution of the spectrometer). The peak centered at 2910 cm$^{-1}$ is assignable to the C–H vibration between carbon and hydrogen nuclei and is thus assignable to the $CH_4$ species. The broad band between 2800 cm$^{-1}$ and 3800 cm$^{-1}$ is assignable to the symmetric stretching vibration of the $H_2O$ species in the liquid or the solid form. The broadband nature of the symmetric stretching vibration of condensed water is due to the existence of a hydrogen bonded network [33–35]. It is well known that the temperature, density, amount of substance of dissolved compounds, as well as the state of the water affect the development of the hydrogen bonded network and, with this, the shape of the respective broad Raman band. The hardly visible marginal differences between the Raman band of pure water (black in Figure 3) before pressurization and the water-rich solution after pressurization (blue in Figure 3) with the gas mixture is due to the dissolved $CO_2$ and $CH_4$ that disturb the hydrogen bonded network. The presence of $CH_4$ and $CO_2$ in the water-rich liquid phase can be extracted from the existence of the $CO_2$- and $CH_4$-characteristic Raman peaks in its spectrum. The Raman spectrum measured from the hydrate gel (red in Figure 3) features a symmetric water stretching band that is significantly different to the Raman spectra of pure water and the water-rich liquid phase at the same temperature. This significant alteration is due to the existence of solid hydrate in the gel, whereby the hydrogen bonded network is much more developed in the solid hydrate than it is in the liquid water. As a consequence, the left shoulder of the broad water stretching vibration band grows in relation to its right shoulder.

Furthermore, it can be seen that the center peak positions of $CO_2$ and $CH_4$ are influenced by the state of phase. When $CO_2$ or $CH_4$ are dissolved in a liquid water-rich phase or incorporated in the cage structure of hydrates, their intramolecular vibrations are influenced by their environment, which causes a peak shift towards smaller Raman shifts.

The ratio of various peak intensities $\frac{I_{CH_4}}{I_{CO_2}}$ within one Raman spectrum:

$$\frac{I_{CH_4}}{I_{CO_2}} = \alpha \frac{n_{CH_4}}{n_{CO_2}} \tag{1}$$

is directly proportional to the ratio of the amount of the respective substances $\frac{n_{CH_4}}{n_{CO_2}}$, with $\alpha$ being the proportionality constant. The intensity $I$ of a peak is computed by fitting model peaks (Gaussian, Lorentzian, or quasi-Voigt profile shapes), as described by Schuster et al. [36] and as frequently applied in the widely spread "Indirect hard modelling" approach [37,38]. Then, the intensity $I$ of the modelled peaks is determined by their integration and taken as the intensity of the real peak.

In order to find the proportionality factor $\alpha$ which is relevant for the applied Raman spectroscopic sensor, we made calibration measurements in each of the available gas mixtures.

Figure 4 shows the correlation of the Raman signal intensity ratio $\frac{I_{CH_4}}{I_{CO_2}}$ and the molar ratio $\frac{n_{CH_4}}{n_{CO_2}}$ in the gaseous binary $CH_4/CO_2$-mixtures. For each composition, ten measurements were repeated. The error bars (see Figure 4) represent the standard deviation of ten repetitions. It has been shown several times that the ratio calibration method is not affected by the state of phase [39,40]. Therefore, a ratio-calibration that has been performed in gaseous mixtures can also be applied to liquids, supercritical fluids, gels, or solids.

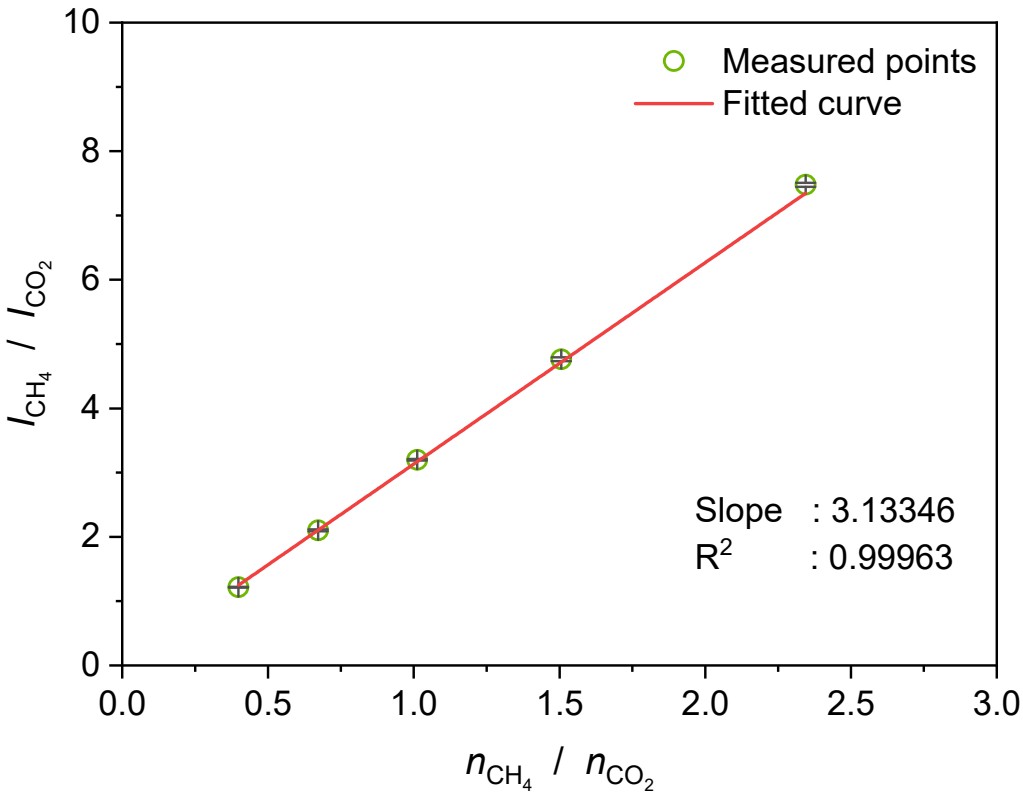

**Figure 4.** The correlation of the Raman signal intensity ratio $\frac{I_{CH_4}}{I_{CO_2}}$ and the amount of substance ratio $\frac{n_{CH_4}}{n_{CO_2}}$ in the gaseous binary $CH_4/CO_2$-mixture.

*3.2. Effect of Feed Gas Composition on the Evolution of $\frac{n^V_{CH_4}}{n^V_{CO_2}}$ in the Vapor Phase*

The temporal evolution of the molar ratio of $CH_4$ and $CO_2$ in the vapor phase $\frac{n^V_{CH_4}}{n^V_{CO_2}}$ is given in Figure 5 for three different initial gas compositions.

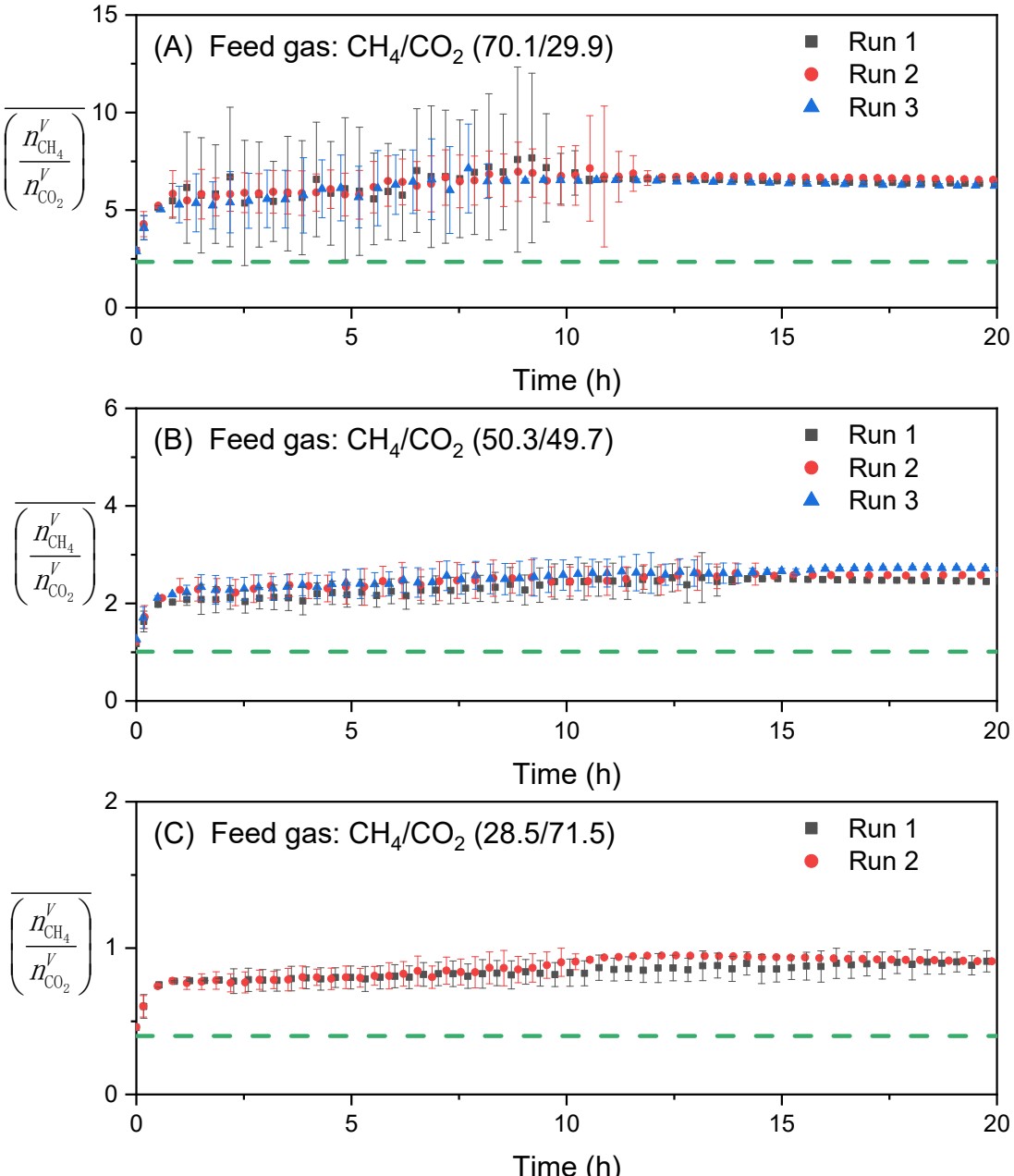

**Figure 5.** The evolution of the molar ratio $\overline{\left(\dfrac{n_{CH_4}^V}{n_{CO_2}^V}\right)}$ in the gaseous phase before and during the hydrate formation at $p_{ini} = 4$ MPa and for three different feed gas compositions: (**A**) methane fraction 0.701, (**B**) methane fraction 0.503 and (**C**) methane fraction 0.285.

The dashed horizontal lines denote the initial $\left(\dfrac{n_{CH_4}^V}{n_{CO_2}^V}\right)^0$ ratio in the feed gas for reference. The colored data points represent mean $\overline{\left(\dfrac{n_{CH_4}^V}{n_{CO_2}^V}\right)}$ values that were averaged from 60 subsequently measured $\dfrac{n_{CH_4}^V}{n_{CO_2}^V}$ values (corresponds to a measurement time of 20 min times 3 spectra per minute). The error bars represent the standard deviation of these 60 measurements. Considering the averaged values together with their standard deviation enables the illustration of several experimental runs in one diagram, as without averaging the diagrams would be overloaded with data points. Each data point train corresponds to one experimental run.

The transition from the dissolution period to the hydrate growth period at time $t_1$ is not extractable (no obvious kink) from the evolutions of $\overline{\left(\frac{n^V_{CH_4}}{n^V_{CO_2}}\right)}$ in the vapor phase, but can be extracted from the temporal pressure profile, as described in the context of Figure 2. It can be seen from Figure 5 that $\overline{\left(\frac{n^V_{CH_4}}{n^V_{CO_2}}\right)}$ at $t_0 = 0$ h is always larger than that of the feed gas. This is due to averaging values with an overall increasing tendency (positive slope) and due to the dissolution of $CH_4$ and $CO_2$ into the liquid water-rich phase already before $t_0$ during the pressurization process. $\overline{\left(\frac{n^V_{CH_4}}{n^V_{CO_2}}\right)}$ increases rapidly within the first 60 min after $t_0$ and slower at later times after $t_1$. The significant variation of $\overline{\left(\frac{n^V_{CH_4}}{n^V_{CO_2}}\right)}$ must therefore be assigned to the dissolution of $CO_2$ and $CH_4$ into the liquid water-rich phase. The gas dissolution period contributes approximately 72.1% ± 0.8%, 71.4% ± 1.0%, and 71.8% ± 3.9% to the total increase of $\overline{\left(\frac{n^V_{CH_4}}{n^V_{CO_2}}\right)}$, respectively. The ratio $\overline{\left(\frac{n^V_{CH_4}}{n^V_{CO_2}}\right)}$ can only increase, if at the same time the ratio $\overline{\left(\frac{n^L_{CH_4}}{n^L_{CO_2}}\right)}$ of the amount of substances of $CH_4$ and $CO_2$ dissolved in the liquid water-rich phase is smaller than $\overline{\left(\frac{n^V_{CH_4}}{n^V_{CO_2}}\right)}$. At the end of the experiment at $t_2$, $\overline{\left(\frac{n^V_{CH_4}}{n^V_{CO_2}}\right)}$ is approximately 2.66 ± 0.08, 2.55 ± 0.13, and 2.31 ± 0.10 times that in the initial feed gas, with methane molar fractions in the feed gas of 0.701, 0.503, and 0.285, respectively.

It is obvious that the error bars are small during the dissolution period, are then first large during the early phase of the hydrate formation period, before later decreasing again. During the early phase of hydrate formation and due to the intensive agitation of the system, liquid water-rich ligaments might precipitate on the windows of the cell or be dispersed into the gaseous phase. These ligaments can convert to hydrate gels and deflect the laser beam or, if they appear close to the probe volume, cause an interference of the desired Raman signals from the gaseous phase and the undesired Raman spectra detected from the ligaments. Once a coherent hydrate gel layer has formed at the interface to the gaseous $CO_2$/$CH_4$-mixture, no more liquid-rich ligaments can be dispersed into the gaseous phase, and thus the error bars become smaller.

### 3.3. Effect of Feed Gas Composition on the Molar Fraction of Hydrate $x^H$ in the Hydrate Gel Phase

As mentioned above, the hydrate gel (superscript G) is composed of pure solid gas hydrate (superscript H) with inclusions of a liquid water-rich phase (superscript L). In order to separate the fraction of water contained in the pure solid hydrate $x^H_{H2O}$ from the fraction of water contained in the included liquid water-rich phase $x^L_{H2O}$ with

$$x^H_{H2O} \;=\; \frac{n^H_{H2O}}{n^H_{H2O} + n^L_{H2O}} \;\&\; x^H_{H2O} \,+\, x^L_{H2O} \;=\; 1 \tag{2}$$

we follow exactly the approach reported by Holzammer et al. [25]. In short, this approach is based on the deconstruction of the broad water stretching vibration Raman band $I^G_{H2O}$ of the gel into contributions assignable to the pure hydrate $I^H_{H2O}$ and assignable to the included liquid water-rich phase $I^L_{H2O}$.

Figure 6 shows the temporal evolution of the fraction of water contained in the hydrate phase $x^H_{H2O}$ for three initial feed gas compositions. $t_1$ is defined by the first data point shown for each experimental run. In general, the curve shapes of the temporal evolutions of $x^H_{H2O}$ are similar to the ones reported by Holzammer et al. [24], but with a deviating final $x^H_{H2O}$. The smallest $x^H_{H2O}$ (0.2434 ± 0.0068) is obtained from the $CH_4$/$CO_2$-mixture with a molar $CH_4$-fraction of 0.701, which is quite similar to the literature

data (0.216) published by Di Profio et al. [41] in a stirred tank reactor at the same initial pressure of 4 MPa and at a temperature of 274 K with a similar molar $CH_4$ fraction of 0.6.

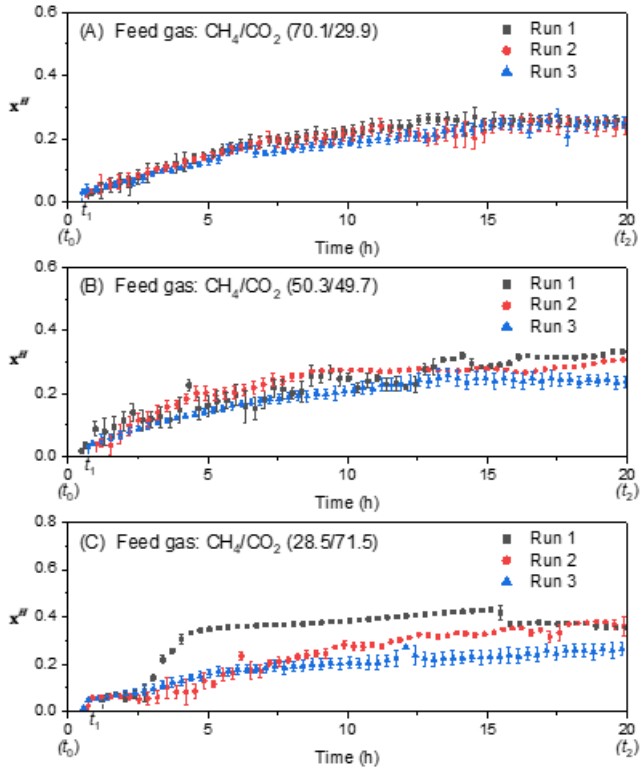

**Figure 6.** The evolution of the fraction of hydrate $x^H$ contained within the gel for three different gas compositions. (**A**) feed gas methane fraction 0.701, (**B**) feed gas methane fraction 0.503 and (**C**) feed gas methane fraction 0.285.

*3.4. Effect of Feed Gas Composition on the Evolution of the Molar Ratio $\frac{n_{CH_4}}{n_{CO_2}}$ in the Liquid Water-Rich Phase $\frac{n_{CH4}^L}{n_{CO_2}^L}$ and in the Solid Hydrate Phase $\left(\overline{\frac{n_{CH_4}^H}{n_{CO_2}^H}}\right)$*

Before hydrate formation (before $t_1$), the molar ratio $\frac{n_{CH4}^L}{n_{CO2}^L}$ of $CH_4$ and $CO_2$ dissolved in the liquid water-rich phase can be computed from:

$$\frac{I_{CH4}^L}{I_{CO2}^L} = \alpha \frac{n_{CH4}^L}{n_{CO2}^L} \tag{3}$$

the ratio of the Raman signal intensities of $I_{CH4}^L$ and $I_{CO2}^L$ with $\alpha = 3.13346$ being the proportionality constant derived during the calibration in various $CH_4/CO_2$ mixtures (see Figure 4). Both $I_{CH4}^L$ and $I_{CO2}^L$ are extractable from the Raman spectra recorded from the water-rich liquid phase. Figure 7 shows the evolution of $\frac{n_{CH4}^L}{n_{CO2}^L}$ in the liquid water-rich phase before $t_1$ (before the hydrate formation). The number of data points per data row vary as the start of the hydrate formation, which is indicated by the kink in the pressure profile, stochastically varies from experiment to experiment. At early times in the dissolution period, the ratios $\frac{n_{CH4}^L}{n_{CO2}^L}$ are small, meaning that significantly more $CO_2$ than $CH_4$ is dissolved into the liquid water-rich phase. Combining this derivation (Figure 7) with the significant pressure decrease obtained especially also at early times in the dissolution period (Figure 2), one can

conclude that the overall change of the vapor phase ratio $\overline{\left(\frac{n^V_{CH_4}}{n^V_{CO_2}}\right)}$ largely depends on the early times of

the dissolution period. With decreasing initial $\left(\frac{n^V_{CH_4}}{n^V_{CO_2}}\right)^0$ in the vapor phase, the ratio $\frac{n^L_{CH4}}{n^L_{CO2}}$ in the liquid

water-rich phase decreases significantly.

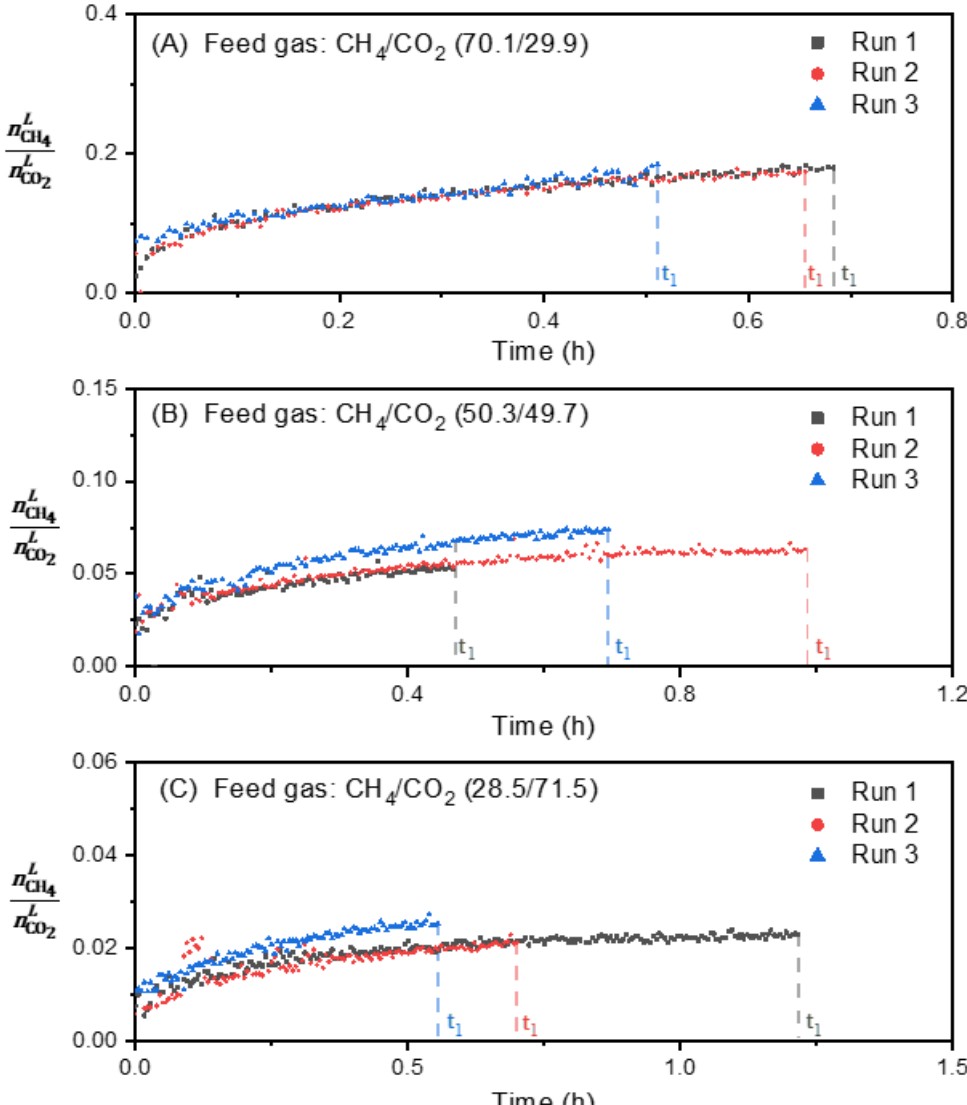

**Figure 7.** The evolution of the molar ratio $\frac{n^L_{CH4}}{n^L_{CO2}}$ in the liquid water-rich phase (before hydrate formation) at $P_{ini} = 4$ MPa and for three different initial gas compositions. (**A**) feed gas methane fraction 0.701, (**B**) feed gas methane fraction 0.503 and (**C**) feed gas methane fraction 0.285.

After hydrate formation, the Raman signals coming from the liquid water-rich phase and from the hydrate phase interfere with each other and together form the Raman signal coming from the hydrate gel. In order to be able to compute the molar ratio $\frac{n^H_{CH4}}{n^H_{CO2}}$ of $CH_4$ to $CO_2$ contained in the pure solid hydrate without the interference from the liquid water-rich phase, we follow the approach by Holzammer et al. [25]. First, we deconstruct the Raman spectrum of the hydrate gel into the intensity contributions assignable to water in the liquid state $I^L_{H2O}$, water in the hydrate state $I^H_{H2O}$, $CH_4$ in the gel $I^G_{CH4}$, and $CO_2$ in the gel $I^G_{CO2}$. These quantities can be extracted directly from the acquired Raman spectra and are written in bold in the following equations.

The Raman signal intensity $I_{CH4}^{G}$ of $CH_4$ in the gel is simply the summation of the $CH_4$ Raman signal intensities coming from the hydrate $I_{CH4}^{H}$ and the liquid phase $I_{CH4}^{L}$. The same is true for $H_2O$ and for $CO_2$. Therefore the ratio

$$\frac{I_{CH4}^{G}}{I_{H2O}^{G}} = \frac{I_{CH4}^{H} + I_{CH4}^{L}}{I_{H2O}^{H} + I_{H2O}^{L}} = \frac{\frac{I_{CH4}^{H}}{I_{H2O}^{H}}}{1 + \frac{I_{H2O}^{L}}{I_{H2O}^{H}}} + \frac{\frac{I_{CH4}^{L}}{I_{H2O}^{L}}}{1 + \frac{I_{H2O}^{H}}{I_{H2O}^{L}}} \tag{4}$$

can be expressed by various other ratios. Assuming that the solubility of $CH_4$ in the liquid water-rich phase remains constant during the hydrate formation after $t_1$, the ratio $\frac{I_{CH4}^{L}}{I_{H2O}^{L}}$ can be extracted from the Raman spectrum recorded just before hydrate formation (just before $t_1$). Thus, $\frac{I_{CH4}^{L}}{I_{H2O}^{L}}$ is also known and in the following is also represented in bold. Consequently, we transform the equation above and obtain an equation for the computation of

$$\frac{I_{CH4}^{H}}{I_{H2O}^{H}} = \left( \frac{I_{CH4}^{G}}{I_{H2O}^{G}} - \frac{\frac{I_{CH4}^{L}}{I_{H2O}^{L}}}{1 + \frac{I_{H2O}^{H}}{I_{H2O}^{L}}} \right) \left( 1 + \frac{I_{H2O}^{L}}{I_{H2O}^{H}} \right). \tag{5}$$

What has been shown above for $CH_4$ can be repeated in an identical manner for $CO_2$, from which

$$\frac{I_{CO2}^{H}}{I_{H2O}^{H}} = \left( \frac{I_{CO2}^{G}}{I_{H2O}^{G}} - \frac{\frac{I_{CO2}^{L}}{I_{H2O}^{L}}}{1 + \frac{I_{H2O}^{H}}{I_{H2O}^{L}}} \right) \left( 1 + \frac{I_{H2O}^{L}}{I_{H2O}^{H}} \right) \tag{6}$$

follows. Additionally, here the ratio $\frac{I_{CO2}^{L}}{I_{H2O}^{L}}$ can be extracted from the last Raman spectrum recorded just before the onset of hydrate formation (just before $t_1$). In a last step, the ratio of the values we obtained for $\frac{I_{CH4}^{H}}{I_{H2O}^{H}}$ according to Equation (5) and $\frac{I_{CO2}^{H}}{I_{H2O}^{H}}$ according to Equation (6),

$$\frac{\frac{I_{CH4}^{H}}{I_{H2O}^{H}}}{\frac{I_{CO2}^{H}}{I_{H2O}^{H}}} = \frac{I_{CH4}^{H}}{I_{CO2}^{H}} = \alpha \frac{n_{CH4}^{H}}{n_{CO2}^{H}} \tag{7}$$

results in the desired molar ratio $\frac{n_{CH4}^{H}}{n_{CO2}^{H}}$ of $CH_4$ and $CO_2$ in the pure hydrate, where $\alpha = 3.13346$ is the proportionality constant we obtained from the calibration measurements in different gas mixtures (see Figure 2).

Figure 8 shows the temporal evolution of the molar ratio of $CH_4$ and $CO_2$ in the pure solid hydrate phase $\left( \frac{n_{CH_4}^{H}}{n_{CO_2}^{H}} \right)$ during the hydrate formation for three different initial gas compositions. The first data point of each data row defines the start of the hydrate formation ($t_1$). The dashed horizontal green lines denote the initial $\left( \frac{n_{CH_4}^{V}}{n_{CO_2}^{V}} \right)^{0}$ ratio in the feed gas for reference. It should be reminded that already after the dissolution period the ratio $\overline{\left( \frac{n_{CH_4}^{V}}{n_{CO_2}^{V}} \right)}$ differs significantly from the initial one (compare Figure 5).

The mean final ratios $\overline{\left( \frac{n_{CH_4}^{V}}{n_{CO_2}^{V}} \right)}$ in the vapor phase for $t_2$ (end of the hydrate formation period) are also provided as reference in Figure 8 (grey horizontal lines). The colored data points represent the mean

$\overline{\left(\dfrac{n_{CH_4}^H}{n_{CO_2}^H}\right)}$ values that were averaged from the 60 subsequently calculated $\dfrac{n_{CH_4}^H}{n_{CO_2}^H}$ values by Equations (5)–(7). The error bars represent the standard deviation of these 60 calculated values. Each data point train corresponds to one experimental run.

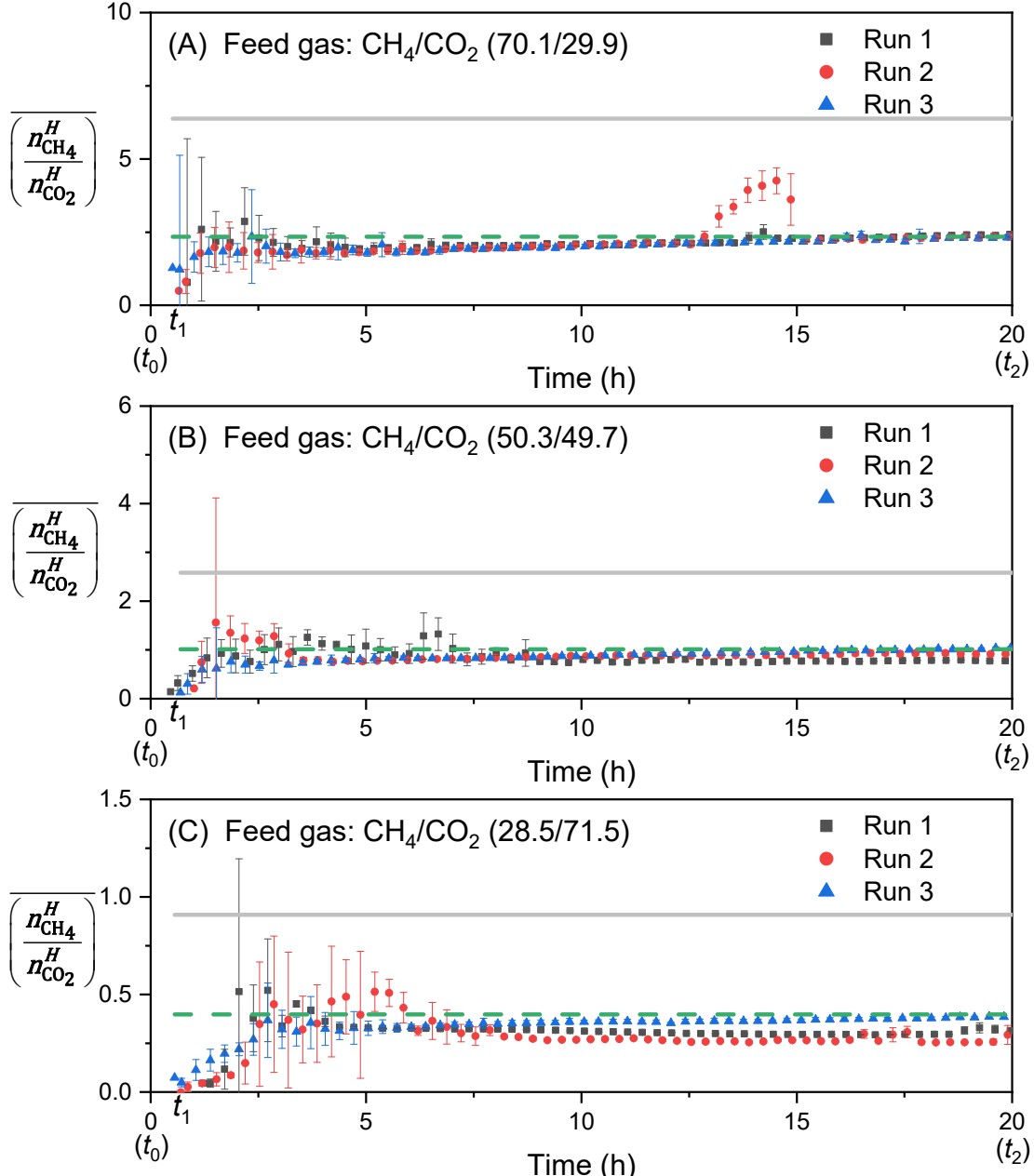

**Figure 8.** The evolution of the molar ratio $\overline{\left(\dfrac{n_{CH_4}^H}{n_{CO_2}^H}\right)}$ in the solid hydrate phase in the gel phase during the hydrate formation at $p_{ini} = 4 MPa$ and for three different initial gas compositions. The dashed lines indicate the initial $CH_4/CO_2$ ratio at $t_0$ of the feed gas, and the solid grey lines indicate the $CH_4/CO_2$ ratio in the vapor phase at the end of the hydrate formation at $t_2$. (**A**) feed gas methane fraction 0.701, (**B**) feed gas methane fraction 0.503 and (**C**) feed gas methane fraction 0.285.

Figure 8 shows that at early times of the hydrate formation period, $\overline{\left(\dfrac{n_{CH_4}^H}{n_{CO_2}^H}\right)}$ starts with small values and later accommodates to a plateau or a marginal linear increase. This means that, initially,

it is mainly $CO_2$ that acts as a guest molecule, while with increasing time $CH_4$ is also consumed for hydrate formation. For all experiments studied, the ratio of $CH_4$ to $CO_2$ in the hydrate phase at $t_2$ is in between the ratios measured in the vapor phase and the one measured in the liquid water-rich phase: $\overline{\left(\dfrac{n^L_{CH_4}}{n^L_{CO_2}}\right)} < \left(\dfrac{n^H_{CH_4}}{n^H_{CO_2}}\right) < \overline{\left(\dfrac{n^V_{CH_4}}{n^V_{CO_2}}\right)}$. Within the first 10 h of the hydrate formation period, some data rows seem to feature a maximum $\overline{\left(\dfrac{n^H_{CH_4}}{n^H_{CO_2}}\right)}$. Unfortunately, these maxima also fall into the regions in which the data points feature rather large error bars. Therefore, their existence can either be justified by a lack in the precision of the measurements made, or their existence might be explained by the circumstance that small hydrate cages ($5^{12}$) in the sI hydrate structure, which prefer $CH_4$ molecules, are more likely to form at the initial stage of the hydrate formation. Both, previously reported experimental studies [42–44] and molecular dynamics simulation studies [45] support this theory. Note that $CO_2$ molecules are not suitable for encaging into the small cavities of sI hydrate because of the molecular size constraint [29]. The peak visible in Run 2 in Figure 8.A between 10 h and 15 h can be explained by a vapor phase bubble that was entrained into the hydrate gel. Still the peak maximum is smaller than the ratio $\overline{\left(\dfrac{n^V_{CH_4}}{n^V_{CO_2}}\right)}$ of the corresponding vapor phase ratio at similar times.

*3.5. Determination of the Selectivity Factor*

The selectivity factor of $CO_2$ ($SF_{CO_2}$) with respect to methane for the liquid water-rich phase at the end of the dissolution period at $t_1$ or for the solid hydrate phase at the end of the hydrate formation period at $t_2$ for three initial gas compositions is defined as

$$SF^i_{co_2} = \frac{\left(\dfrac{n^i_{CO_2}}{n^i_{CH_4}}\right)}{\left(\dfrac{n^V_{CO_2}}{n^V_{CH_4}}\right)} = \frac{\dfrac{1}{\left(\dfrac{n^i_{CH_4}}{n^i_{CO_2}}\right)}}{\dfrac{1}{\left(\dfrac{n^V_{CH_4}}{n^V_{CO_2}}\right)}} \tag{8}$$

where the superscript $i$ could be either L or H, which denotes the liquid water-rich phase or the pure solid hydrate phase, respectively. Selectivity factor of $CO_2$ $SF^L_{co_2}$ for the liquid water-rich phase just before $t_1$ are determined as 29.27, 33.72, and 33.2 for the initial molar $CH_4$-fractions in the feed gas of 0.701, 0.503, and 0.285, respectively. Similarly, $SF^H_{co_2}$ for the solid hydrate phase are 2.70, 2.81, and 2.86, respectively. The results show that $SF^H_{co_2}$ is almost the same despite varying the initial gas compositions. Moreover, $SF^L_{co_2}$ is larger than $SF^H_{co_2}$ at each initial gas composition.

The selectivity factor of $CO_2$ ($SF^G_{co_2}$) for the hydrate gel phase, including the $CO_2$ dissolved in the liquid-water-rich phase and the $CO_2$ captured in the solid hydrate, was calculated using the conventional method based on the amount of gas consumed for the entire reaction process. The detailed description of this method can be found elsewhere in previous studies [41,46]. The gas compositions at $t_2$ in the vapor phase used in the calculation could be converted from the mole ratiosis taken from the data shown in Figure 5. $SF^G_{co_2}$ for the hydrate gel phase at $t_2$ are determined as 6.4, 8.1, and 8.2 for the initial molar $CH_4$-fractions in the feed gas of 0.701, 0.503, and 0.285, respectively. These values are in agreement with those reported by Di Profio et al. [41] from 3.1 to 7.1 and Zhong et al. [46] from 6.5 to 11.2. Also the selectivity factor of $CO_2$ ($SF_{CO_2}$) follows the order $SF^H_{co_2} < SF^G_{co_2} < SF^L_{co_2}$. Therefore, it seems that the selectivity factor of $CO_2$ for hydrate might be overestimated in previous studies with the gas consumption method, if one considered using $SF^G_{co_2}$ instead of $SF^H_{co_2}$ to judge the performance of a separation process by forming hydrate.

## 4. Conclusions

Within this study we investigated into the suitability of a gas hydrate formation process for the separation of $CH_4$ and $CO_2$ from a feed gas or at least for the purification of one of the two compounds. One of the main findings of this study is that the main separation is achieved not by hydrate formation but by the preferential absorption of $CO_2$ into the liquid water-rich phase before the onset of hydrate formation. Also the hydrate phase preferentially (with respect to the vapor phase composition) incorporates $CO_2$, and therefore in principle could contribute to a further purification of $CH_4$ in the remaining vapor phase. But as the fraction of liquid water converted into solid hydrate is rather small with less than 30 mol%, the separation achieved by hydrate formation is small compared to the separation achieved by absorption. This can become significantly different, if more solid hydrate is formed. More solid hydrate can be formed if for example a more efficient mixing between the vapor and the gel phase is assured or if either the liquid water-rich phase is atomized through the vapor phase or the vapor phase is dispersed through the continuous liquid water-rich phase.

**Author Contributions:** Conceptualization, Z.L., C.C.H., and A.S.B.; data curation and analysis, Z.L. and C.C.H.; writing—original draft preparation, Z.L.; writing—review and editing, A.S.B.; supervision, C.C.H. and A.S.B.; funding acquisition, A.S.B. All authors have read and agreed to the published version of the manuscript.

**Funding:** The project leading to this contribution has received funding from the European Union's Horizon 2020 research and innovation programme under ERC Starting Grant agreement No. 637654 (Inhomogeneities). The authors also gratefully acknowledge the funding from China Scholarship Council (Grant No. 201806050175) for supporting the visit of Zheng Li at TU Bergakademie Freiberg.

**Conflicts of Interest:** The authors declare no competing financial interest.

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
