# Peer review of "Analysis of the Dissolution of CH4/CO2-Mixtures into Liquid Water and the Subsequent Hydrate Formation via In Situ Raman Spectroscopy"

_energies, doi:10.3390/en13040793_

Round 1
Reviewer 1 Report
The manuscript presents new data of the CH4/CO2 composition of the vapor, aqueous and hydrate phase during dissolution and hydrate formation by Raman spectroscopy. I recommend that the work is published subject to minor revisions.
Line 41-44: Do these gels exist or persist in systems which do not have an excess of water? Line 342 – 345: This might be the case, but have the authors considered the molar ratio of water to gas that has been used (29 to 32.5 moles of water per mole of gas) is much higher than the approximately 6 typical of gas hydrates. This means that the system had an excess of water meaning that the system will never convert all the water to hydrate – even if the pressure remained sufficient. For example, for the methane + ethane system this has been considered in Figure 4 of Subramanian et al [1] (all regions of the diagram have liquid water present). Hughes and Marsh [2] considers the differences between having excess gas and excess water. The separation factor will change also with the molar ratio of water to gas. I suggest the authors consider highlight that their measurements were completed with excess water in their manuscript. Also, does the system reach equilibrium in the time of the experiments? Given that the initial composition of the vapor and water added to the system and the final composition of the vapor phase, liquid water rich phase and hydrate phase as well as the fraction of hydrate are known, a material balance should be completed as a consistency check of the measured compositions. Are there any measurements of solubility of CO2 and CH4 in water that the dissolution measurements can be compared to? The materials and methods section should be written in the past tense. Line 52: suggest replacing the word “manifold”. I am not sure what was meant by this sentence. Line 73: suggest replacing “for launching” with “to pass”. Line 77: Was it a PT100 (a platinum resistance temperature sensor) or a thermocouple? They are different things. Line 116: suggest replacing “subtracted” with “removed” Line 121: “as” should be “a” Fig 4: suggest using non-filled symbols so that error bars can be clearly seen when printed. Fig 5: suggest mole fractions rather than mole ratios are plotted. Also dashed lines were almost invisible when printed. Fig 5: Why was it necessary to average 60 points of data? Fig 5 A and B – why do the error bars get so much smaller after 12 or 13 hours? Fig 7: suggest mole fractions rather than mole ratios are plotted. Line 270: I don’t agree with the use of “directly measureable quantity”. Surely the data deconstruction means that these quantities are not directly measureable.References
[1] S. Subramanian, A.L. Ballard, R.A. Kini, S.F. Dec, E.D. Sloan, Structural Transitions in Methane + Ethane Gas Hydrates. Part I: Upper Transition Point and Applications, Chem. Eng. Sci., 55 (2000) 5763-5771.
[2] T.J. Hughes, K.N. Marsh, Measurement and Modeling of Hydrate Composition During Formation of Clathrate Hydrate from Gas Mixtures, Ind. Eng. Chem. Res., 50 (2011) 694-700.
Author Response
The authors are very grateful to the reviewer for his/her comments and suggestions and very much appreciate the time and the commitment the reviewer invested into our manuscript. Below we comment point by point the suggestions/comments of the reviewer, which we followed.
Reviewer 1
Line 41-44: Do these gels exist or persist in systems which do not have an excess of water?
According to experience of the authors gels or slurries can also form from systems that do not feature an excess of water. It depends on the dimensions of the systems, how intensively it is stirred and how much time is provided for the slow mass transfer of water and CO2 through the hydrate layer that forms at the interface between the liquid and the gaseous phase. The authors are afraid that this is a rather complex systems and do not want to comment on this in their manuscript. The uptake of the gas compounds into the hydrate phase also reduces the pressure in the system which again has a huge impact onto the driving force to form more hydrate.
Line 342 – 345: This might be the case, but have the authors considered the molar ratio of water to gas that has been used (29 to 32.5 moles of water per mole of gas) is much higher than the approximately 6 typical of gas hydrates. This means that the system had an excess of water meaning that the system will never convert all the water to hydrate – even if the pressure remained sufficient. For example, for the methane + ethane system this has been considered in Figure 4 of Subramanian et al [1] (all regions of the diagram have liquid water present). Hughes and Marsh [2] considers the differences between having excess gas and excess water. The separation factor will change also with the molar ratio of water to gas. I suggest the authors consider highlight that their measurements were completed with excess water in their manuscript.
In the revised version it is highlighted that an excess of water was used and that usually other ratios of gas to water are used (last sentence subsection 2.3):
“Considering the operational conditions, the molar ratio n_H2O⁄((n_CO2+n_CH4)) fed into the cell is larger than 50. In studies of other groups this ratio is often set close to 6 [Ref]. We chose a rather large ratio in order to obtain a significant change of the composition in the gas phase coexisting to the hydrate slurry/gel.”
Also, does the system reach equilibrium in the time of the experiments? Given that the initial composition of the vapor and water added to the system and the final composition of the vapor phase, liquid water rich phase and hydrate phase as well as the fraction of hydrate are known, a material balance should be completed as a consistency check of the measured compositions.
Mass balance: Unfortunately this measuremet procedure described here only measures the ratio of CH4 to CO2 or the ratio of liquid water to solid water. The quantity that would be required to close the mass balance was the ratio of CH4 to water or CO2 to water. These quantities we did not measure.
Thermodynamics: Using the software provided by the hydrate research center at Colorado school of mines we computed the pressure to be reached in thermodynamic equilibrium and compared it with the pressure we reached at t2. Yes, the theoretical phase equilibrium pressures calculated with each CH4/CO2 compositions measured in the vapor phase at t2 were almost the same with the pressures at t2. For example, at 276K, the calculated pressure at t2 was 2.20 MPa while the measured pressure was ~ 2.16 MPa. We commented on this in the revised version of the manuscript (lines 142-146):
“The pressure 2.16 MPa reached at t_2 is close to the pressure of 2.2 MPa expected for thermodynamic equilibrium according to the software CSMHYD/GEM, developed by the hydrate research center of the Colorado School of Mines. The hydrate layer formed at the interface to the gaseous phase acts as mass transfer resistance and kinetically can limit further hydrate growth and with this also the pressure drop towards t_2.”
Are there any measurements of solubility of CO2 and CH4 in water that the dissolution measurements can be compared to?
Here we did not measure the solubility of methane or CO2 in water, but only the ratio of both compounds in either the gaseous phase, the liquid phase or the hydrate phase. Solubility measurements can be found in the works of Holzammer et al that we cited in the revised version ()also cited in the original submission)
The materials and methods section should be written in the past tense.
Corrected to the past tense. Thanks
Line 52: suggest replacing the word “manifold”. I am not sure what was meant by this sentence.
We deleted the word “manifold” accordingly.
Line 77: Was it a PT100 (a platinum resistance temperature sensor) or a thermocouple? They are different things.
Sorry for this mistake. We changed to PT100 resistance temperature sensor.
Line 116: suggest replacing “subtracted” with “removed”
We have changed the word “subtracted” with “removed” in the manuscript according to the reviewers’ suggestion.
Line 121: “as” should be “a”
We modified the sentence to make the meaning of the sentence clear.
Fig 4: suggest using non-filled symbols so that error bars can be clearly seen when printed.
Thanks for the reviewers’ comments. We corrected Figure 4 with non-filled symbols accordingly.
Fig 5: suggest mole fractions rather than mole ratios are plotted. Also dashed lines were almost invisible when printed.
Thanks for the reviewers’ comments. The results in this study were all based on the measured Raman spectra. And the ratios of different species in the system were directly related to the ratios of intensities subtracted from the Raman spectrum. Thus, we believe it would be more clear for the readers using the mole ratios rather than mole fractions in Figure 5. Also we corrected the color (green) of the dashed lines in Figure 5 to make sure they can be clearly seen when printed.
Fig 5: Why was it necessary to average 60 points of data?
There were too many (thousands of) data points in every experiment run as we obtained one Raman spectrum per second. We justified the averaging in the revised version as follows (lines 227 to 229):
“. Considering averaged values together with their standard deviation enables the illustration of several experimental runs in one diagram, as without averaging the diagrams would be overloaded with data points.”
Fig 5 A and B – why do the error bars get so much smaller after 12 or 13 hours?
We added this explanation in the revised version (lines 245 to 253):
“It is obvious that the error bars are small during the dissolution period, then are first large during the early phase of hydrate formation period and later decrease again. During the early phase of hydrate formation and due to the intensive agitation of the system liquid water-rich ligaments might precipitate on the windows of the cell or be dispersed into the gaseous phase. These ligaments can convert to hydrate gels and deflect the laser beam or, if they appear close to the probe volume, cause an interference of the desired Raman signals from the gaseous phase and the undesired Raman spectra detected from the ligaments. Once a coherent hydrate gel layer has formed at the interface to the gaseous CO2/CH4-mixture, no more liquid-rich ligaments can be dispersed into the gaseous phase and thus the error bars become smaller.”
Fig 7: suggest mole fractions rather than mole ratios are plotted.
See previous comment to previous similar suggestion
Line 270: I don’t agree with the use of “directly measureable quantity”. Surely the data deconstruction means that these quantities are not directly measureable.
We have deleted the sentence of “directly measureable quantities” in the revised manuscript.
References
[1] S. Subramanian, A.L. Ballard, R.A. Kini, S.F. Dec, E.D. Sloan, Structural Transitions in Methane + Ethane Gas Hydrates. Part I: Upper Transition Point and Applications, Chem. Eng. Sci., 55 (2000) 5763-5771.
[2] T.J. Hughes, K.N. Marsh, Measurement and Modeling of Hydrate Composition During Formation of Clathrate Hydrate from Gas Mixtures, Ind. Eng. Chem. Res., 50 (2011) 694-700.
Reviewer 2 Report
The authors have investigated the CH4 + CO2 hydrate formation process for gas separation with Raman spectroscopy. These results are important to understand the gas molecule enclathration phenomena during the hydrate formation. And the mechanism is helpful to develop gas separation processes. However, the deconvolution of Raman spectra and the discussions about the selectivity factor were not adequately explained. Therefore, I recommend that this manuscript is accepted for publication after the authors consider the following points:
P. 1
The results should be explained before the discussion about the main separation effect. And important things with regard to gas separation with the hydrate in this paper should be described at the end.
P. 3, L. 44-50
Issues of the previous study should be explained with the contents of the previous study. For example, the comparison of CO2 composition, gas consumption or gas separation factor in previous studies and these theoretical values are shown as evidence of the error. If you show the results, readers would understand the significance of the errors.
P. 5 L. 124, P. 6, Figure 2
Why were the solid line and the dashed lines discriminated? In Figures 6 - 8 (C), there were the t1 in either Ran. You should correct the sentence “t1 only shown for the solid black curve).
P. 5 L. 127
It was explained that the cause of the pressure kink was a temperature spike with the exothermic of the hydrate formation. However, the amount of hydrate formation was a little at t1. I think that the hydrate formation started after the amount of gases dissolution reached the saturation solubility.
P. 6, Figure 2
Figure of the pressure as a function of time should be shown separately from the position of the Raman prove. The lines and t1, t2 should be explained in figure caption of the pressure as a function of time. Figure of the position of the Raman prove and the explanation about Raman measurements should be shown in “2.3 Procedures”.
P. 7, L. 159-161
Raman peaks of gaseous CO2 and gaseous CH4 are different from that of CO2 and CH4 in the hydrate cages. This knowledge should be explained because it is necessary that the Raman peaks corresponding to CO2 and CH4 molecules were deconvoluted with fitting models.
P. 8, L. 184, Figure 4
In reference 25 relating to co-author of this paper, there was the sentence “the intensity (integrated peak area)”. The intensity I of a peak is different from the integrated peak area. If the intensity I in this paper was the integrated peak area, typical results of deconvolution in the vapor phase and the hydrate gel phase should be shown. In that case, definition of the intensity I would be clear. For your information, gaseous CO2 is known to have major bands and minor bands. The CH4 molecules enclathrated in small cage (512) and large cage (51262) appears as two distinct hands.
P. 9, Figure 5, P. 14, Figure 8
Color of the dashed lines should be deepened further because I can hardly see these lines.
P. 11, Figure 6
Could you explain about the cause of the variation of the fraction of hydrate in Run 1 in Figure 6 (C) between 5 h and 15 h? Was the hydrate gel fluidized state? Was the Raman peak affected by an attached hydrate gel on the window and an aggregated hydrate gel at an interface between vapor and liquid phase?
I recommend that the theoretical hydrate fraction is estimated from an amount of the gas consumption after t1 and the injected water, because the comparison to the theoretical value would improve a reliability of the result of the hydrate function.
P. 15
Selectivity factor of CO2 for gel phase is important for a gas separation process with a hydrate. And then a contribution of selectivity factor of CO2 for liquid phase to that for gel phase is shown. In addition, the selectivity factor for gel phase could be compared to some literature values for the selectivity factor, and the authors should discuss about the selectivity factor.
P. 15, L. 340, 341
What was the theoretical phase equilibrium pressure? If the pressure at t2 was near the phase equilibrium pressure, the hydrate fraction would not increase. It seems to me that how to increase a hydrate fraction is not important in this paper. If a hydrate fraction relates to the selectivity factor for solid hydrate phase, the authors should discuss about that.
Author Response
The authors are very grateful to the reviewer for his/her comments and suggestions and very much appreciate the time and the commitment the reviewer invested into our manuscript. Below we comment point by point the suggestions/comments of the reviewer, which we followed.
Reviewer 2
The authors have investigated the CH4 + CO2 hydrate formation process for gas separation with Raman spectroscopy. These results are important to understand the gas molecule enclathration phenomena during the hydrate formation. And the mechanism is helpful to develop gas separation processes. However, the deconvolution of Raman spectra and the discussions about the selectivity factor were not adequately explained. Therefore, I recommend that this manuscript is accepted for publication after the authors consider the following points:
1
The results should be explained before the discussion about the main separation effect. And important things with regard to gas separation with the hydrate in this paper should be described at the end.
In the revised version of the abstract we changed the arrangement according to the suggestion of the reviewer. Thank you very much.
3, L. 44-50Issues of the previous study should be explained with the contents of the previous study. For example, the comparison of CO2 composition, gas consumption or gas separation factor in previous studies and these theoretical values are shown as evidence of the error. If you show the results, readers would understand the significance of the errors.
Thanks for the reviewers’ comments. The comparison of the results in this work and in the literature was added in the revised manuscript. (see Page 20, Lines 340-350)
5 L. 124, P. 6, Figure 2
Why were the solid line and the dashed lines discriminated? In Figures 6 - 8 (C), there were the t1 in either Ran. You should correct the sentence “t1 only shown for the solid black curve).
The solid line and the two dashed lines in Figure 2 represent three independent experimental runs performed at the same conditions. Here we only intend to show the specific times t1 or t2 indicating the different period of experiment (end of gas dissolution period or end of experiments). So one t1 for one experiment is enough. While in Figures 6-8, we show each t1 in either experimental run because the trend curves start at different t1 for each corresponding experimental run. To make it clearer we added “instant” in front of t1.
5 L. 127
It was explained that the cause of the pressure kink was a temperature spike with the exothermic of the hydrate formation. However, the amount of hydrate formation was a little at t1. I think that the hydrate formation started after the amount of gases dissolution reached the saturation solubility.
In the revised version we quantified the temperature increase (spike) in line 137
6, Figure 2
Figure of the pressure as a function of time should be shown separately from the position of the Raman prove. The lines and t1, t2 should be explained in figure caption of the pressure as a function of time. Figure of the position of the Raman prove and the explanation about Raman measurements should be shown in “2.3 Procedures”.
We put the pressure profiles together with the sketch of the position of Raman probe in Figure 2 in order to show how we obtain the Raman spectra in different state of phase as a function of time. According to the reviewers’ suggestion, we added several sentences to explain the lines and t1, t2 in the figure caption in the revised manuscript (see Page 6, Lines 149-152).
7, L. 159-161
Raman peaks of gaseous CO2 and gaseous CH4 are different from that of CO2 and CH4 in the hydrate cages. This knowledge should be explained because it is necessary that the Raman peaks corresponding to CO2 and CH4 molecules were deconvoluted with fitting models.
In fact we have explained the Raman peaks of CO2 and CH4 in different phase in Figure 3 and text in the manuscript (Page 7, context figure 3). We also showed the difference of central peak positions affected by the state of phase (Page 7, context figure 3).
8, L. 184, Figure 4
In reference 25 relating to co-author of this paper, there was the sentence “the intensity (integrated peak area)”. The intensity I of a peak is different from the integrated peak area. If the intensity I in this paper was the integrated peak area, typical results of deconvolution in the vapor phase and the hydrate gel phase should be shown. In that case, definition of the intensity I would be clear. For your information, gaseous CO2 is known to have major bands and minor bands. The CH4 molecules enclathrated in small cage (512) and large cage (51262) appears as two distinct hands.
The intensity I in this paper was always the integrated peak area. The deconvolution methods to acquire the intensities for different species were described in detail in the literature (Schuster, J. J.; et al. J Raman Spectrosc 2014, 45 (3), 246-252.). Unfortunately, we cannot distinguish CH4 molecules enclathrated in small cage (512) and large cage (51262) appears as two distinct peaks due to the limited resolution (~15 cm-1) of the Raman spectrometer we used in this work. Therefore we added in line 176:
“(additional CO2 peaks merged with the CO2 Fermi Dyad, due to the resolution of the spectrometer)”
9, Figure 5, P. 14, Figure 8
Color of the dashed lines should be deepened further because I can hardly see these lines.
Thanks for the reviewers’ comments. We corrected the color (green) of the dashed lines to make sure they can be clearly seen when printed.
11, Figure 6
Could you explain about the cause of the variation of the fraction of hydrate in Run 1 in Figure 6 (C) between 5 h and 15 h? Was the hydrate gel fluidized state? Was the Raman peak affected by an attached hydrate gel on the window and an aggregated hydrate gel at an interface between vapor and liquid phase?
I recommend that the theoretical hydrate fraction is estimated from an amount of the gas consumption after t1 and the injected water, because the comparison to the theoretical value would improve a reliability of the result of the hydrate function.
If just by accident hydrate accumulated in regions of the probe volume that is not moved and mixed with the bulk water, the end values can be measured already earlier.
15Selectivity factor of CO2 for gel phase is important for a gas separation process with a hydrate. And then a contribution of selectivity factor of CO2 for liquid phase to that for gel phase is shown. In addition, the selectivity factor for gel phase could be compared to some literature values for the selectivity factor, and the authors should discuss about the selectivity factor.
Thanks for the reviewers’ comments. We added a paragraph to explain the selectivity factor of CO2 for the hydrate gel phase and compared the results with the literature data. Also we discussed the selectivity factor in the revised manuscript. (last part before the Conclusion section)
15, L. 340, 341
What was the theoretical phase equilibrium pressure? If the pressure at t2 was near the phase equilibrium pressure, the hydrate fraction would not increase. It seems to me that how to increase a hydrate fraction is not important in this paper. If a hydrate fraction relates to the selectivity factor for solid hydrate phase, the authors should discuss about that.
The theoretical phase equilibrium pressures calculated with each CH4/CO2 compositions measured in the vapor phase at t2 were almost the same with the pressures at t2. For example, at 276K, the calculated pressure at t2 was 2.20 MPa while the measure pressure was ~ 2.16 MPa. Sure the hydrate fraction seems will not dramatically increase at the experimental conditions in the manuscript. But if the hydrate gel converts completely to the solid hydrate the situation would be totally different, as we have showed in the manuscript that the selectivity factor of CO2 for the liquid water-rich phase is much higher than that for the solid hydrate phase .